# Integrating Microorganism-Based Therapy and Emerging Biotechnology in the Treatment of Intracranial Central Nervous System Diseases

**DOI:** 10.3390/pharmaceutics17091175

**Published:** 2025-09-09

**Authors:** Zifan Li, Shihua Yang, Lida Su

**Affiliations:** 1The Second Affiliated Hospital, College of Medicine, Zhejiang University, Hangzhou 310009, China; lizf24@zju.edu.cn; 2Department of Breast Surgery, Cancer Hospital of Dalian University of Technology, Shenyang 110042, China

**Keywords:** intracranial central nervous system, blood-brain barrier, microorganism-based drug delivery system, emerging biomedical technology, clinical trials

## Abstract

The development of drug delivery systems for the treatment of intracranial central nervous system (CNS) diseases remains one of the most intractable medical problems in modern society, owing to the special physiological structure of the brain, including the existence of the blood-brain barrier (BBB), the CNS’s immune privilege, and its high complexity and vulnerability. Recently, a leading approach in the CNS drug delivery domain has been to employ or simulate the physiological behavior of microorganisms to overcome the BBB and remodel the pathological immune microenvironment in intracranial tissue. Considering the exceptional advancements in microorganism-based CNS drug delivery systems, it is imperative to review the latest breakthroughs. Herein, we summarize the emerging trends at the intersection of microorganism-based drug delivery systems and emerging biomedical technology for the treatment of CNS diseases, with a particular focus on preclinical research into microorganism-based drug delivery systems to combat CNS diseases, aiming to describe a credible landscape for further clinical trials.

## 1. Introduction

Undoubtedly, the CNS is one of the most important and complex systems in the human body; it not only controls and regulates all fundamental bodily functions but also plays a crucial role in cognition, emotion, and behavior. However, the clinical treatment of CNS diseases is extremely challenging [1,2]. First, as a highly selective and protective barrier, the BBB obstructs the entry of various drugs into the brain, such as paclitaxel and cisplatin for the treatment of glioblastoma, donepezil and vilazodone for Alzheimer’s disease, acyclovir for meningitis, and levodopa for Parkinson’s disease [3]. In addition, the immune privilege of the CNS results in insufficient immune responses when the brain faces challenges such as infections, tumors, and neurodegenerative diseases, leading to poor therapeutic outcomes [4]. This characteristic poses a significant challenge to the treatment of CNS disorders, particularly in the areas of immunotherapy and drug delivery. To date, drug delivery systems used for the clinical treatment of CNS disorders have struggled to effectively overcome both of these obstacles simultaneously [5]. Therefore, it is essential to rationally design and develop novel drug delivery systems that can successfully penetrate the BBB and modulate the intracranial immune environment.

Recently, various novel drug delivery systems have been developed for the treatment of CNS diseases, including transcytosis-based fusion protein, nanoparticles, and exosomes [2,6]. For example, research on delivering a fusion protein consisting of the lysosomal enzyme iduronate 2-sulfatase (IDS) and an anti-human transferrin receptor antibody (JR-141) into the CNS for treating Hunter syndrome has progressed to a phase II clinical trial [7]. A study on targeting glutathione transporters at the BBB by conjugating glutathione to the surface of doxorubicin liposomes yielded satisfactory results in a phase I clinical trial [8]. In addition, a study demonstrated that in a zebrafish model of brain cancer, exosomes carrying the anticancer drugs doxorubicin and paclitaxel were successfully detected in the brain after intravenous injection, whereas free drugs were not [9]. Intravenous injection of dopamine-loaded exosomes significantly increased intracranial dopamine levels [10]. However, due to high investment costs, poor efficacy, and significant individual variability, these drug delivery systems have not progressed beyond the clinical research stage to enter the market.

Researchers have recently discovered that various microorganisms can successfully enter the intracranial space and alter the intracranial immune microenvironment. Given this discovery, a growing body of research has focused on microorganism-based therapies for the treatment of CNS disorders [11]. First, various natural microorganisms, such as K1 bacteria and adeno-associated viruses (AAVs), can efficiently cross the BBB, making them effective drug delivery vehicles for transporting therapeutics into the brain to exert their effects [12]. Furthermore, owing to the intrinsic immunogenicity of microorganisms, microorganism-based CNS drug delivery systems can actively regulate immune responses and reconstruct the pathological intracranial immune microenvironment, offering unique advantages over other therapeutic approaches [13]. Moreover, microorganism-based drug delivery systems provide ample capacity for loading various therapeutic agents for the treatment of CNS diseases, especially excelling in the delivery of DNA/RNA drugs in recent years, which significantly enhances the accumulation of such drugs within the brain. However, as foreign entities, microorganisms possess significant immunogenicity and self-replication capabilities. If their fate within the body is not properly controlled, they may cause intracranial immune dysregulation, posing a serious threat to the host organism. Therefore, it is worth delving into research on how to safely and efficiently utilize microbial delivery systems to transport drugs to the intracranial region.

There is no doubt that progressive biomedical innovations hold a huge probability to broaden a splendid prospect for the emerging treatments of diseases [14,15]. Notably, many emerging biomedical technologies are expanding possibilities for the development and application of microorganism-based drug delivery systems against CNS diseases by enhancing the safety and efficacy of these systems [15]. For instance, loading therapeutic microbial drugs into stem cells, such as neural and mesenchymal stem cells, harnesses the safety and intracranial targeting properties of these cells, facilitating the successful delivery of microbial drugs to intracranial lesions, thereby exerting therapeutic effects [16]. Furthermore, microorganism-based gene editing technology has greatly facilitated the treatment and in-depth exploration of the mechanisms underlying CNS diseases, such as Alzheimer’s disease and stroke [17]. In addition, by using genetic engineering techniques to knock out the surface toxic proteins of bacteria that have the ability to cross the BBB and extracting bacterial outer membrane vesicles to replace bacteria as drug delivery carriers, researchers can achieve a dual guarantee to avoid the negative impact of immunogenicity on the intracranial central nervous system [18]. Overall, the integration of microorganism-based drug delivery systems and emerging biomedical technology presents a unique opportunity to advance the precision treatment of CNS diseases.

Considering the rapid development of microorganism-based drug delivery systems against CNS diseases, it is timely to comprehensively review the latest advances in this emerging field. Herein, the CNS barrier’s physiological structure, the foundation of common CNS diseases, and the cross-talk between the pathogenesis of CNS diseases and microorganisms are outlined. Furthermore, the emerging biomedical technologies used for microorganism-based drug delivery systems against CNS diseases are summarized in detail. Last but not least, we emphasize the clinical expectations and challenges of integrating microorganism-based therapy and emerging biotechnology in the treatment of intracranial central nervous system diseases.

## 2. Foundation of Microorganism-Based CNS Disease Therapeutics

### 2.1. CNS Barriers and Anatomy

The maintenance of homeostasis within the CNS is essential for neuronal and brain function and plays a key part in ensuring the integrity of the organism and a harmonious balance with the external environment [19]. The presence of the BBB plays a crucial role in maintaining homeostasis. The BBB consists of the cerebral capillary endothelial barrier, the blood-cerebrospinal fluid barrier, the blood–retina barrier, the blood-ependymal barrier, and the blood-arachnoid barrier. As a multicellular and dynamic interface, it can strictly control the passage of various nutrients and obstruct the penetration of harmful xenobiotic molecules. The main part of blood vessels is composed of tightly connected endothelial cells, limiting the diffusion of cells and molecules through the endothelial space into the brain parenchyma. Endothelial cells also provide a physical barrier to the gap around blood vessels by producing a basement membrane composed of laminin α4 and laminin α5. In addition, pericytes, a type of cell that surrounds endothelial cells in capillaries and veins throughout the body, can maintain the stability of the blood–brain barrier during development. Furthermore, microglia, which are closely juxtaposed with the vascular endothelium, are the inherent immune effector cells in the CNS and are involved in a series of neurodegenerative diseases. The activation of microglia and neuroinflammation are the main features of neuropathology [20]. The maintenance of BBB stabilization also depends on the formation of glial boundaries and a dense network of basal lamina [21]. Overall, the functions of various cells in the CNS complement each other and participate in the construction of the BBB and the maintenance of CNS homeostasis.

### 2.2. Outline of CNS Diseases

CNS diseases remain a severe threat to human health, as reported in the Global Burden of Disease study (GBD) [22]. The obscure pathogenesis, immanent immune privilege, and difficult drug passage doubtlessly pose non-negligible challenges for disease treatment and postoperative recovery. In this section, we intend to outline common CNS diseases with a focus on their pathogenesis, treatment strategies, and clinical situations.

#### 2.2.1. Intracranial Cancer

Intracranial tumor undoubtedly represents one of the most intractable CNS diseases. In contrast to other cancers, even benign intracranial tumors will elicit different degrees of functional damage once compressed to any site of the brain, not to mention advanced tumors or even diffuse malignancies [23]. According to its pathogenesis, intracranial cancer can be generally divided into primary and metastatic types [24]. Primary intracranial cancer is most commonly found in children, whereas metastatic intracranial cancer is usually discerned in multiple patient populations and includes liver, breast, and lung cancers [25]. In addition to well-known triggering factors, such as genetics and the environment, viral infections can also induce intracranial tumors. For example, studies have shown that human cytomegalovirus infection is associated with the occurrence of intracranial medulloblastoma [26]. JC virus, as an oncogenic virus, can infect oligodendrocytes and astrocytes in the CNS, thereby inducing intracranial cancers [27]. The mainstream clinical treatments of intracranial tumors are still surgery, chemotherapy, and radiotherapy. Although various emerging immunotherapy approaches, such as oncolytic virus therapy (OVT), immune checkpoint inhibitors (ICI), and CAR-T therapy, have successively entered the stage of clinical research, no related products have appeared on the market [28]. There is an ascending trend in the incidence of intracranial tumors, which account for about 5% of total tumor cases and 70% of childhood tumor cases [29]. Therefore, how to comprehensively improve the postoperative life quality and prolong the lifespan of patients is undoubtedly the crux of intracranial cancer clinical research.

#### 2.2.2. CNS Inflammation

CNS inflammation represents one of the brain diseases most closely related to microbial invasion. CNS inflammation can be divided into acute encephalitis and chronic neuroinflammation. Common types of encephalitis include epidemic, sporadic, and autoimmune encephalitis. The pathogenesis of encephalitis is mostly due to viruses and bacteria. Typically, some COVID-19 patients have symptoms of encephalitis [30]. CNS inflammation involves the activation of microglia and astrocytes, which release pro-inflammatory cytokines and chemokines, leading to the recruitment of immune cells to the site of injury. Key molecular pathways, such as the NF-κB, MAPK, and JAK-STAT pathways, play crucial roles in regulating the inflammatory response and contribute to neuronal damage and dysfunction. The clinical treatment of acute encephalitis involves the use of antivirals, antibiotics, corticosteroids, or other drugs, depending on the cause of the inflammation. Chronic neuroinflammation is an immune response activated by microglia and astrocytes in the CNS, which is often associated with the progression of neurodegenerative diseases such as Alzheimer’s disease, Parkinson’s disease, amyotrophic lateral sclerosis, and multiple sclerosis [31]. In addition to factors such as microbial and physical damage, the aging of brain cells is also a major factor leading to the onset of chronic neuroinflammation.

#### 2.2.3. Neurodegenerative Diseases

Neurodegenerative diseases are a class of disorders characterized by the gradual loss of function or death of neurons in the nervous system, typically leading to a decline in cognitive and motor function [32]. These diseases are mostly chronic and progressive, impacting the quality of life of patients. Based on the primary systems or functions affected, neurodegenerative diseases can be classified into diseases of the motor nervous system (e.g., Parkinson’s disease, amyotrophic lateral sclerosis, and Huntington’s disease), cognitive diseases (e.g., Alzheimer’s disease), and mixed neurodegenerative diseases (e.g., Lewy body dementia and multiple system atrophy) [33]. The specific etiology of neurodegenerative diseases is often complex and multifactorial, primarily involving genetics and the environment. In addition, microbial triggers also represent a main factor. For example, certain pathogens can cause brain infections, such as viral encephalitis or viral meningitis, which may damage the nervous system, leading to neuronal injury and neurological dysfunction. In addition, chronic infections induced by microorganisms may lead to neuronal damage by triggering inflammation and promoting abnormal protein aggregation [34]. Research indicates that herpes simplex virus type 1 (HSV-1), human immunodeficiency virus (HIV), cytomegalovirus (CMV), Epstein-Barr virus (EBV), and varicella zoster virus can all induce abnormal aggregation of β-amyloid and tau proteins. Furthermore, infections such as *Porphyromonas gingivalis* and *Helicobacter pylori* can also cause abnormal aggregation of β-amyloid and tau proteins, thereby triggering Alzheimer’s disease [35]. In addition, abnormal gut microbiota composition may lead to inflammatory responses and the abnormal release of metabolic products, which could impact the function of the nervous system and be associated with the development of neurodegenerative diseases [36]. The treatment of neurodegenerative diseases typically includes pharmacotherapy, rehabilitation training, supportive therapy, and lifestyle adjustments. Pharmacotherapy regimens should be planned based on specific etiology and symptoms, and include neurotransmitter modulators (levodopa), anti-inflammatory drugs, and anti-amyloid drugs (Aducanumab) [37]. In conclusion, there is an association between microbiota and the pathogenesis of neurodegenerative diseases. A comprehensive consideration of this relationship may contribute to the development of more effective intervention measures for the prevention and treatment of neurodegenerative diseases.

#### 2.2.4. Stroke

Stroke is a common acute cerebrovascular disease, mainly caused by the sudden blockage or rupture of cerebral arteries, resulting in insufficient blood supply to the brain, leading to cerebral tissue ischemia, hypoxia, and consequent functional impairment [38]. Stroke can be classified into two major types: ischemic and hemorrhagic. Ischemic stroke accounts for approximately 80–85% of all stroke cases, with major etiologies including atherosclerosis, thrombus formation, arterial stenosis, or occlusion. On the other hand, hemorrhagic stroke accounts for about 15–20% of all stroke cases, with primary causes being hypertension-related subarachnoid hemorrhage, intracerebral hemorrhage, and other conditions leading to vascular rupture and bleeding. Furthermore, in recent years, there have been reports showing a certain mutual influence and association between microorganisms and stroke [39]. For example, gut microbiota may impact the pathogenesis of stroke through the modulation of the immune system, effects on metabolic products, and the production of neurotransmitters [40]. Bacteria and viruses in the oral cavity can spread to other sites through the bloodstream, triggering inflammatory reactions, exacerbating diseases like atherosclerosis, and consequently increasing the risk of stroke [41]. Similarly, microbial infections can trigger inflammatory responses, which may lead to pathological processes such as atherosclerosis and thrombus formation, making stroke more likely to occur [42]. Additionally, some studies suggest that prolonged or excessive use of antibiotics may disrupt the balance of gut microbiota, leading to dysbiosis, which in turn affects the host’s immune function and inflammatory status, thereby also increasing stroke risk [43]. In terms of clinical management, acute treatment for stroke includes thrombolytic therapy, anticoagulant therapy, and surgical interventions, aiming to restore cerebral blood flow and minimize brain damage. Chronic phase management mainly consists of rehabilitation training, pharmacological interventions (warfarin, clopidogrel, and donepezil), and lifestyle modifications, aiming to enhance patients’ quality of life and reduce the risk of recurrence [44]. Overall, stroke is a severe cerebrovascular disease that significantly impacts patients’ health and quality of life.

### 2.3. Cross-Talk Between CNS Diseases and Microorganisms

The presence of the BBB creates a highly secure biological environment for the CNS. The complicated dynamic interface plays a key role in maintaining CNS homeostasis by rigorously performing a “gate-controlled switch” to control the passage of specific nutrients while blocking the entry of exogenous ingredients, typically microorganisms [3]. Even so, an increasing number of studies affirmed the relationship between the CNS and microorganisms from multilateral perspectives. First, studies have shown that microorganisms can realize remote communication with the CNS through the gut-brain axis: (1) Microorganisms play a key role in CNS development and BBB formation. (2) With a “regulator” standing, they exert leverage to mediate the onset and progression of multiple CNS diseases, including Alzheimer’s disease, autism spectrum disorder, multiple sclerosis, Parkinson’s disease, and stroke [36].

Except for non-contacting communication, it has been reported that many pathogenic microorganisms can traverse the BBB and enter the intracranial CNS by virtue of their own unique physiological structure. For example, as the cause of suppurative meningitis in newborns, Gram-negative *Escherichia coli* K1 (EC-K1) possesses the ability to cross the BBB via a receptor-ligand binding mechanism [45,46]. Specifically, there exists outer membrane protein A (OmpA) at the surface of EC-K1, which can combine with gp96 protein expressed by BBB endothelial cells, thereby mediating the transcytosis of EC-K1 [45,47]. Except for the classic ompA-gp96 pathway, there are other relevant proteins that are associated with the process of EC-K1 crossing the BBB, typically the CNF1 protein [48]. Moreover, it has been reported by Wang et al. that *Streptococcus pneumoniae*, *group B Streptococcus*, and *neonatal meningitis Escherichia coli* can cross the BBB by hijacking the iron transporter receptor of BBB endothelial cells, resulting in bacterial meningitis and cerebral palsy [49]. Isaac Chiu et al. furthermore probed the pathogenesis of bacterial meningitis, claiming that pathogens such as *Streptococcus pneumoniae* not only cross the membrane by hijacking the relative protein but also release signaling molecules to block the aggregation of immune cells [50]. In addition, Horst Schroten et al. found that *S. agalactiae* can invade brain microvascular endothelial cells (BMECs) by relying on lamini-binding protein (Lmb), fibrinogen-binding protein (FbsA), pili, and invasion-associated gene A (IagA) [51]. Heinz et al. reported that *L. monocytogenes* combines with BMECs by internalin B (InlB) [52]. *S. pneumoniae* arrives at the brain based on the interaction between cell-wall phosphorylcholine and the platelet-activating-factor (PAF) receptor [53]. The process of *N. meningitidis* is related to the recognition between protein Opc and fibronectin [54]. In addition to the above-mentioned encephalitis-causing bacteria, there are many other pathogenic bacteria that can also cross the BBB and enter the CNS, such as *M. tuberculosis* [55].

Unlike the above bacteria that cross the BBB relying on receptor-ligand recognition, there are many studies that show that bacteria can enter the CNS via other methods. Typically, the invasion process of *S. pneumoniae* is promoted by increasing the level of tumor necrosis factor (TNF)-α due to the upregulation of *S. pneumoniae* phosphorylcholine relative receptor [56]. Transforming growth factor (TGF)-β1 plays an arresting role in the invasion of *E. coli* K1 [57]. Moreover, Trojan Horse camouflage is also a common strategy for bacteria to invade the CNS [58] (Figure 1).

In addition to bacteria, the invasion of viruses can also induce a variety of CNS diseases [59]. For instance, there are multiple authoritative reports that indicate that the occurrence of multiple sclerosis is associated with Epstein-Barr virus, and researchers argue that the pathogenesis of multiple sclerosis may be directly related to the combination of Epstein-Barr virus protein (EBNA1) and a specific brain protein, i.e., GlialCAM [59,60].

There are three main pathways by which viruses enter the CNS: (1) by crossing the vascular endothelium; (2) by invading the peripheral nervous system; and (3) by simulating the Trojan horse [61]. First, similar to bacteria, invading vascular endothelium cells is also a main way for viruses to enter the CNS. Many viral invasion processes rely on the special receptor protein on the surface of vascular endothelium. For example, reoviruses can recognize the junctional protein junctional adhesion molecule A (JAM-A), a signature protein on the surface of the vascular endothelium that participates in the formation of tight inter-endothelial barriers [62]. Retrovirus HTLV1 possesses the capability of invading the BBB by recognizing and connecting receptor proteins, including glucose transporter type 1 (GLUT), heparin sulfate proteoglycans, and neuropilin-1 [63]. In addition, many viruses can regulate the expression of some proteins through the upregulation of chemokines, so as to promote the viral leapfrog process. Recently, it has been reported that SARS-CoV-2 can degrade the basilar membrane by upregulating the MMP9 protein and subsequently crossing the BBB via the transcellular pathway [64]. Polyomavirus, a virus that is listed in the 2B carcinogen group by the World Health Organization’s International Agency for Research on Cancer, can realize BBB invasion, relying on serotonin receptor 2A [65]. Mouse adenovirus type 1 (MAV-1) can express the BBB-destroying E3 protein and enter the brain to cause acute encephalomyelitis [66]. Matrix metalloproteinase-9 (MMP-9) can mediate BBB destruction triggered by the West Nile virus, thereby leading to severe encephalitis [67].

Another way in which viruses invade the CNS is by infecting peripheral sensory and motor neurons. For instance, poliovirus can bind to the poliovirus receptors of neurons (CD155). Adenovirus and coxsackieviruses rely on adenovirus receptors (CAR). Acetylcholine receptors and nerve cell adhesion molecules (NCAMs) mediate rabies virus infection [68]. In addition, nectin-1 (also known as PVRL1) and nectin-2 (PVRL2) can act as targets for herpes virus and pseudorabies virus to infect sensory neurons [69]. It has also been reported that, in addition to sensory and motor neurons, the olfactory nerve acts as a target for viruses entering the CNS [70].

Furthermore, viruses also invade the BBB vascular endothelium cells through the Trojan horse mechanism. Typically, some viruses infect immune cells and travel with them across the BBB. For example, lentiviruses and human immunodeficiency viruses usually migrate across the vascular barrier of the CNS by infecting monocytes and macrophages [71,72] (Figure 2). Intriguingly, researchers have found a novel mechanism for the invasion process of viruses: an enzyme called CA-IV (carbonic anhydrase IV) that enables a number of different viral vectors to cross the BBB [73].

In addition to bacteria and viruses, other microorganisms are also closely associated with CNS diseases. For example, Ekberg et al. reported that chlamydia pneumoniae, a Gram-negative respiratory pathogen, can invade the brain through the nasal nerves and cause Alzheimer’s disease [74]. Overall, all of the above-mentioned studies prove the relationship between the occurrence of CNS diseases and microorganisms. The cross-talk between the treatment of CNS diseases and microorganisms will be introduced in the next section (Table 1).

## 3. Emerging Microorganism-Based Therapy for Intracranial Cancer Treatment

### 3.1. Integrating Virus-Associated Therapy and Emerging Biotechnology Against Intracranial Cancer

The COVID-19 pandemic has made the world more aware of the existence of viruses. As organisms that coexist with humans in the ecosystem, viruses play an important role in the occurrence and development of human diseases.

#### 3.1.1. Integrating Oncolytic Virus into Cell-Based Technologies Against Intracranial Cancer

Oncolytic viruses (OVs) are a type of virus characterized by their specific selectivity in infecting and lysing tumor cells without invading normal cells [75]. Common OVs include adenovirus, herpes simplex virus, measles virus, reovirus, and vaccinia virus. These common types of OVs play crucial roles in cancer therapy, with the specific selection depending on factors such as tumor type, patient condition, and treatment objectives [76]. In recent years, the study of OVs in the treatment of intracranial tumors has attracted much attention. The advantages of OVs in the field of intracranial tumors treatment are as follows: First, OVs possess the characteristic of selectively infecting tumor cells, with relatively fewer infections of normal cells, thereby reducing damage to healthy tissues [77]. Furthermore, OVs have multiple anti-tumor mechanisms. Specifically, OVs not only directly infect and lyse tumor cells but may also enhance anti-tumor effects by activating the body’s immune system, leading to a comprehensive therapeutic outcome. Research even suggests that OVs can further exert anti-tumor effects by disrupting tumor vasculature. In addition, OVs exhibit high genetic engineering versatility, allowing for adjustments based on specific needs to enhance their selective infection and cytotoxicity towards tumor cells or to introduce specific gene payloads for achieving more precise therapeutic effects [78]. Based on the aforementioned advantages, OV immunotherapy has garnered significant attention in recent years. However, simple OVs do not have the ability to cross the BBB and are recognized and cleared by immune cells and neutralizing antibodies in the body before they reach the tumor site, which greatly limits their clinical application in the treatment of intracranial tumors [79]. Therefore, integrating emerging biomedical technologies into OV immunotherapy to achieve precise tumor targeting and effective treatment for intracranial tumors is of paramount importance.

As one of the most promising biomedical technologies, cell therapy utilizes living cells or their products for the treatment of diseases. In the medical field, cell therapy has been widely applied to various types of disease treatment [80]. For example, CAR-T cell therapy has achieved significant success in the treatment of leukemia and lymphoma [81]. Stem cell therapy has been utilized for the repair of cardiac tissue damage, assisting patients with heart disease in recovering cardiac function. Encouragingly, there are many studies integrating OVs into cell therapy against intracranial cancer and making progress in clinical studies [82]. As the first cell carrier approved by the Food and Drug Administration for brain tumor therapy, neural stem cells (NSCs) also demonstrate significant advantages in delivering oncolytic viruses for the treatment of brain diseases, with clinical trials currently underway [83]. Atique U Ahmed et al. loaded a glioma-restricted OV CRAd-S-pk7 into NSCs and comprehensively assessed its capabilities against glioblastoma compared to simple CRAd-S-pk7 [84]. Benefiting from the inherent tumor-tropic migratory ability, the capacity to cross the BBB, and the excellent biocompatibility of NSCs, CRAd-S-pk7 loaded into NSCs demonstrated an excellent ability to inhibit tumor growth in a human glioma orthotopic xenograft model, with the most meaningful outcome being an increase in median survival by approximately 50%. Furthermore, the results also show that OV infection upregulates different chemoattractant receptors and significantly enhances the migratory capacity of NSCs both in vitro and in vivo. Subsequently, the author further investigated the in vivo fate of CRAd-S-pk7 loaded by NSCs after administration [85]. Analogously, MA Tyler et al. tried to infect NSCs with the CRAd-S-pk7 virus and discovered that compared to the simple OV control group, the experimental group exhibited superior therapeutic effects in the U87MG intracranial glioma model [86]. Building upon the preliminary research findings, Atique U Ahmed et al. further investigated the synergetic effects of NSCs carrying OVs with other common clinical approaches for the treatment of glioblastoma [87]. Concretely, first, the authors loaded the oncolytic virus CRAd-S-pk7 into NSCs (HB1.F3-CD). Subsequently, they attempted to combine it with ionizing radiation (XRT) and temozolomide (TMZ), meticulously examining the replicative potency of OVs under combined application conditions and the tumor therapeutic capabilities of the combined therapy. The experimental results demonstrate that the combination of XRT and TMZ did not negatively impact the replicative capacity and tumor-killing ability of CRAd-S-pk7. Compared to monotherapy with oncolytic viruses alone, the combined application effectively enhances anti-tumor growth capabilities and can extend the median survival time of mice with glioblastoma by approximately 46%. Interestingly, it was found that the timing and sequence of treatment implementation also have an influence on treatment effectiveness.

In addition to intracranial injection in the above-mentioned studies, related research on administration through the nasal mucosa has also been reported. Mahua Dey et al. found that NSCs can migrate to the brain tumor site through intranasal delivery, and hypoxic preconditioning or overexpression of CXCR4 significantly enhances the tumor-targeting ability of NSCs. To achieve this, the authors increased the surface expression of CXCR4 on NSCs through hypoxic preconditioning and genetic modification and used these engineered NSCs to encapsulate OVs. Subsequently, they administered the OVs loaded in NSCs via the nasal mucosa route to investigate the therapeutic effects on intracranial glioma [88]. The experimental results indicate that hypoxia-treated NSCs demonstrate enhanced intracranial targeting capability, which is associated with the high expression of CXCR4. Loading OVs onto these engineered NSCs shows exceptional tumor therapeutic efficacy. Additionally, it is noteworthy that compared to invasive intracranial injection, nasal mucosa administration is suitable for repeated treatments, which better aligns with clinical treatment needs. However, intranasal drug delivery also has significant drawbacks, such as rapid clearance of the drug in the nasal passage. The research of Drew Spencer et al. provides a reasonable solution for this problem [89]. Concretely, the authors pretreated glioma-bearing mice with methimazole (MT) before intranasally administering NSCs loaded with OVs. The experimental results indicate that MT can disrupt the olfactory epithelial cells, effectively reducing the clearance rate of NSCs, allowing them to persist in the nasal cavity for at least 24 h after a single dose, thereby increasing the efficiency of NSCs reaching the intracranial space and exerting anti-tumor effects.

In addition to NSCs, mesenchymal stem cells (MSCs) are also commonly used to load OVs to treat intracranial tumors. Raymund L. Yong et al. selected an OV (Δ24-RGD) as a model drug that possesses selective infectivity towards glioma cells [90]. In their research, Δ24-RGD was loaded into MSCs (hMSC-Δ24) and injected into the carotid artery of mice bearing orthotopic U87MG or U251-V121 tumor. The experimental results showed that compared to the control group, mice treated with hMSC-Δ24 had a significantly suppressed tumor growth rate and an increased survival rate, which indicates that hMSC-Δ24, after intracarotid injection, can successfully cross the BBB, target tumor tissue, and ultimately achieve cytotoxic effects on them.

The OV delivery system based on stem cell therapy not only significantly inhibits the growth of gliomas but also shows therapeutic effects on other types of brain tumors. Wanlu Du et al. constructed an intracranial delivery system based on MSCs loaded with oncolytic herpes simplex virus (oHSV) as the model drug (MSC-oHSV) [16]. In their study, utilizing mice with melanoma brain metastases as the experimental model, MSC-oHSV and purified oHSV were administered through carotid artery injection. The experimental results indicated that compared to the purified oHSV group, MSC-oHSV could more efficiently target metastatic lesions in the brain, leading to a significant extension of survival in mice treated with MSC-oHSV. Furthermore, following the combined treatment of MSC-oHSV with the immune checkpoint inhibitor PD-L1 antibody, there was a significant increase in the number of IFNγ^+^CD8^+^ tumor-infiltrating T lymphocytes at the intracranial lesions in mice, leading to a notable extension of survival. Overall, stem cells serve as ideal delivery vehicles for OVs in intracranial delivery systems. First, the low immunogenicity of stem cells helps evade the rapid clearance of OVs by neutralizing antibodies and immune cells during the delivery process. Second, stem cells can carry OVs across the BBB and possess tumor tropism, providing a convenient way for OVs to infect tumor cells. Additionally, stem cells offer a site for the replication of OVs, enabling the amplification of the therapeutic dose during the delivery process.

In addition to stem cell therapy, there are also cases where OVs are combined with other cell therapies to enhance treatment effectiveness against CNS tumors. For example, Liu et al. constructed an engineered OV in combination with CAR-T therapy to achieve synergistic treatment of glioblastoma [91]. Considering the important role of IFN-γ in tumor cell killing and the recruitment of tumor-infiltrating cells, the authors designed an HSV capable of transforming malignant tumor cells into IFN-γ factories and combined it with CAR-T cells. The experimental results showed that the upregulation of IFNγ levels in the tumor microenvironment not only effectively recruited CAR-T cells, thus enhancing anti-tumor efficacy, but also reshaped the suppressive tumor immune microenvironment, which was characterized by a decrease in the proportion of regulatory T cells and a reduction in the level of transforming growth factor β.

#### 3.1.2. Integrating OVs into Membrane Encapsulation Technology Against Intracranial Cancer

Membrane encapsulation technology refers to the biomimetic modification of a cell membrane structure on the surface of a drug delivery system to improve its properties, such as biocompatibility, safety, and tumor selectivity. It represents one of the most promising emerging technologies in the field of drug delivery [92]. For example, Nie et al. fused *Escherichia coli* membranes with tumor cell membranes and used this hybrid membrane structure to encapsulate PLGA nanoparticles, synergistically enhancing the anti-tumor activity of the tumor vaccine [92]. Huang et al. screened the optimal formulation ratio of erythrocyte membranes and liposome membranes and used it to encapsulate OVs to prolong their circulation time after administration and enhance anti-tumor activity [93].

Research on delivering OVs to the intracranial space for the treatment of CNS diseases by means of membrane encapsulation technology has also attracted attention. Considering that stem cells as carriers cannot efficiently deliver OVs to the brain through intravenous administration and are instead sequestered in the liver, Yu and colleagues tried to reform the intracranial OV delivery system [94]. The authors first designed and developed OVs (A4/k37) with selective infectivity for glioblastomas. To enable A4/k37 to cross the BBB via intravenous delivery and target glioblastoma cells, the authors encapsulated A4/k37 with neural stem cell (NSC) membranes and glioblastoma cell membranes (NCM-Ad and GCM-Ad). The experimental results indicate that, compared to the simple A4/k37 administration group, both NCM-Ad and GCM-Ad demonstrate superior abilities to cross the BBB and target tumors in vitro and in vivo, thereby more effectively inducing tumor cell apoptosis through the caspase8–caspase3 pathway (Figure 3).

#### 3.1.3. Integrating OVs into Gene Modification Technology Against Intracranial Cancer

Gene modification technology is undoubtedly one of the most impactful technologies in the field of drug delivery [95]. Gene modification technology can reform the properties of drug delivery systems in multiple dimensions, including improving pharmacokinetic properties, increasing targeting specificity, enhancing drug stability, reducing drug toxicity, and synergistically enhancing efficacy. The fact that OVs inherently carry a large amount of genetic material provides a convenient condition for gene modification. A large number of preclinical and clinical studies of OVs involve gene modification technology. For example, Wang et al. designed a novel oncolytic adenovirus (Ad) by inserting a fragment of the IL-12 [96]. A novel OV modified by inserting granulocyte-macrophage colony-stimulating factor into an oncolytic poxvirus has entered clinical stage I [97].

Based on gene modification technology, Yu et al. designed and constructed a dual-engineered OV for the treatment of glioblastoma [98] (Figure 4). Concretely, the secretable single-chain variable fragment of the epidermal growth factor receptor (EGFR) antibody cetuximab and the gene segment of C-C motif ligand 5 (CCL5) were introduced into HSV (OV-Cmab-CCL5). Upon reaching intracranial tumor tissue, the EGFR antibody cetuximab will recognize and bind to the highly expressed EGFR on the surface of glioblastoma cells, mediating the entry of OV-Cmab-CCL5 into tumor cells and producing a large amount of CCL5. Both in vitro and in vivo experimental results demonstrate that treatment with OV-Cmab-CCL5 significantly enhances the infiltration and activation of natural killer cells, macrophages, and T cells, effectively suppressing tumor growth and prolonging the survival time of tumor-bearing mice. It has been reported that tumor cells increase the expression of cellular interferon-inducible genes after viral infection, leading to the activation of the double-stranded RNA-dependent host protein kinase (PKR), which inhibits the translation efficiency of host cell peptides, thereby limiting intracellular viral replication. Kevin A. Cassady et al. developed a novel OV using gene modification technology that can block the generation of PKR, thereby enhancing the replication of the virus [99].

Montserrat Puigdelloses et al. constructed a novel engineered OV based on Delta-24-RGD, named Delta-24-ACT, which is capable of expressing 4-1BB ligand (4-1BBL), and evaluated its therapeutic effect on glioblastoma by means of immunofluorescence, flow cytometry, and other techniques [100]. The experimental results showed that Delta-24-ACT successfully infected human brain glioblastoma cell lines in vitro and completed replication within host cells, ultimately leading to cell lysis. Additionally, the functional 4-1BBL expressed by Delta-24-ACT was able to co-stimulate T lymphocytes in vitro and in vivo, thereby reshaping the suppressive immune microenvironment at the tumor site. In addition, compared to Delta-24-RGD, Delta-24-ACT demonstrated stronger anti-tumor effects during the treatment of glioblastoma in mice, significantly extending the survival period of the experimental group mice. However, prolonged stimulation by OVs led to an increased expression of programmed cell death protein 1 (PD-1) on T cells, resulting in the suppression of anti-tumor efficacy. To address this problem, the authors attempted to combine Delta-24-ACT with an anti-PD-L1 antibody. In vivo pharmacological results demonstrated that the combination group exhibited a more significant therapeutic effect than any single therapy alone and could induce immune memory to inhibit tumor recurrence.

#### 3.1.4. Integrating OVs into Other Biomedical Technologies Against Intracranial Cancer

Various biomedical technologies are employed in applications of OV-based intracranial delivery systems for cancer treatment. M.M. Alonso et al. observed that an OV (DNX-2401) showed certain therapeutic efficacy in the clinical treatment of pediatric diffuse intrinsic pontine glioma (DIPG). To obtain the latest relevant clinical research data, DNX-2401 was injected into the patients’ brains and subsequently combined with radiotherapy. Clinical data show that among the 12 patients in the treatment group, 9 survived for over 12 months, including 3 individuals who survived for over 24 months. Overall, the intratumoral infusion of DNX-2401 combined with radiotherapy is feasible for treating DIPG in clinical stages [101].

The immunogenicity of OVs is the main obstacle to their systemic delivery, primarily due to the rapid clearance of OVs by preexisting IgM in the body. To address this, KEIRO IKEDA et al. attempted to combine OVs with cyclophosphamide (CPA), which inhibits this innate antiviral response by reducing the overall plasma IgM levels [102]. The experimental results showed that the combination of CPA significantly increased the concentration of intracranial OVs, indicating that more OVs successfully crossed the BBB to enter the brain. Additionally, in a mouse model of intracranial glioma, the combination group exhibited better therapeutic efficacy (Table 2).

#### 3.1.5. Virus-Based Gene Delivery Technologies Against Intracranial Cancer

The gene drug delivery system represents a novel technology that introduces gene drugs into desired cells through vectors, regulating the expression of specific proteins [104]. In recent years, significant progress has been made in the field of treatment of various diseases, such as malignancies, metabolic diseases, immune disorders, and neurodegenerative diseases [105]. Commonly used carriers for gene drug delivery include viral vectors, lipid nanoparticles, extracellular vesicles, and polymer-based carriers. Among them, viruses as gene delivery carriers have advantages such as high efficiency, specificity, long-term expression, wide applicability, and engineering modification capability, making them one of the most commonly used and effective carriers in gene therapy.

As a secreted protein of the TNF superfamily, tumor necrosis factor ligand superfamily member 14 (TNFSF14, LIGHT, and CD258) has been widely used in cancer therapy, owing to its ability to induce tumor-associated high endothelial venules (HEVs) and tertiary lymphoid structures (TLSs), thereby facilitating the recruitment and activation of T cells [106]. For this purpose, Anna Dimberg and colleagues designed and constructed an adeno-associated virus (AAV) vector carrying the LIGHT (AAV-LIGHT) gene with brain endothelial cell-targeting capability for the treatment of glioblastoma [107]. The experimental results show that post-administration, AAV-LIGHT can successfully target and alter the shape and function of tumor vasculature with the help of the expression of LIGHT, providing favorable conditions for recruiting cytotoxic T cells against the tumor. Furthermore, AAV-LIGHT plays a key role in orchestrating the formation of immune cell aggregates associated with brain tumors, known as tertiary lymphoid structures (TLSs), and is crucial for the regulation of cell types within these aggregates, primarily characterized by the upregulation of the proportion of T cells. Additionally, this engineered AAV demonstrates the ability to recruit stem-like T cells, which is crucial for the activation of specific anti-tumor immunity and for reshaping the immunosuppressive microenvironment within gliomas. The pharmacological results indicate that the systemic delivery of AAV-LIGHT can significantly inhibit the growth rate of tumors in a mouse model of glioblastoma resistant to anti-PD-1 monoclonal antibody therapy and prolong their survival. Furthermore, immunological results demonstrate that treatment with AAV-LIGHT promotes the recruitment of stem-like T cells, reduces T cell exhaustion, and relies on enhanced immune memory to prevent tumor recurrence. In general, this preclinical study on virus-based gene therapy for anti-PD-1 monoclonal antibody-resistant glioblastoma holds profound significance for the exploration of subsequent clinical trials. Similarly, Duane Mitchell et al. developed a novel AAV expressing C-X-C motif ligand 9 (CXCL9) to address the clinical issue of insensitivity to PD-1 immune checkpoint inhibitors in many glioblastoma patients. The experimental results demonstrated that the engineered AAV, upon reaching the target site and being taken up by glioblastoma cells, significantly upregulated the expression of CXCL9, which facilitated the recruitment of CD8^+^ T cells to the tumor site, rendering GBM more sensitive to anti-PD-1 immune checkpoint blockade (ICB) [108]. To address the strong immunosuppressive nature of the glioblastoma microenvironment, M C Burger et al. developed a combination immunotherapy [109]. Specifically, the authors first engineered a surface-presented ankyrin repeat protein with the ability to target human epidermal growth factor receptor, into which the mRNA of aPD-1 was introduced (HER2-AAV). Subsequently, the authors combined this engineered AAV with anti-HER2.CAR/NK-92 cells, which could block the recognition of HER2-AAV. After co-administration, activated anti-HER2.CAR/NK-92 cells altered surrounding tumor cells and bystander immune cells by inducing the release of inflammatory cytokines and the upregulation of PD-L1. Subsequently, HER2-AAV selectively infected tumor cells through specific recognition by DARPin, completing the translation of aPD-1 mRNA within host cells, resulting in high intratumoral and low systemic levels of aPD-1 expression, hence improving safety during treatment. The experimental results showed that significantly increased intratumoral concentrations of aPD-1 could be detected up to 10 days following a single injection of HER2-AAV. In addition, compared to the control group, the combination therapy exhibited the best tumor suppression efficiency and significantly prolonged the survival of mice. Considering the important roles of reactive astrocytes and tumor-associated macrophages/microglia in the formation and progression of gliomas, Adrienn Volak et al. developed a novel AAV that specifically targets and infects microglia and reactive astrocytes. This AAV carries a gene expressing the potent anti-tumor cytokine interferon β. The experimental results demonstrate that compared to the control group, the tumor growth in the experimental group of mice was significantly inhibited, leading to a notable extension in survival rates [110].

To further enhance the ability of AAV vectors to cross the BBB, Fengfeng Bei et al. developed a novel AAV variant (AAV.CPP.16), which demonstrated superior delivery efficiency compared to the control group in preclinical animal experiments [111] (Figure 5). Concretely, the researchers screened hundreds of cell membrane-penetrating peptides with a series of tests and continuous optimizations and ultimately found that AAV.CPP.16 possessed the optimal ability to cross the BBB. Subsequently, the authors conducted investigations into different species of animal models. The results showed that, compared to the commonly used AAV9, AAV.CPP.16 exhibited significantly enhanced delivery efficiency across the blood-brain barrier: an approximately 6-fold increase in brain infection efficiency in different mouse strains and an increase of around 5-fold in the cynomolgus macaque model. Subsequently, the authors investigated the specificity and safety of AAV.CPP.16, and the experimental group exhibited outstanding affinity for infecting neurons, astrocytes, and spinal motor neurons in the brain. Although it also possessed some infectivity towards cells in other organs, it did not induce serious diseases such as significant and sustained liver toxicity reactions or degeneration of the dorsal root ganglia. Furthermore, the authors explored the potential of AAV.CPP.16 as a gene therapy vector for the treatment of malignant gliomas. Specifically, leveraging AAV.CPP.16, the authors designed and constructed a gene delivery system capable of expressing immune checkpoint inhibitors. The pharmacological results indicated that, compared to AAV9, AAV.CPP.16 significantly enhanced the infiltration of CD8^+^ T cells and led to a marked reduction in the proportion of Treg cells at the tumor site, demonstrating a satisfactory anti-tumor efficacy. In addition, the authors further utilized AAV.CPP.16 for the delivery of the herpes simplex virus thymidine kinase gene, in combination with ganciclovir. The experimental results demonstrated a synergistic anti-tumor effect, a reduction in tumor volume, and an increase in the survival rate of tumor-bearing mice to approximately 57%. Similarly, Viviana Gradinaru et al. identified two AAV vectors (AAV.CAP-B10 and AAV.CAP-B22) with efficient BBB crossing capabilities using the same strategy. This discovery laid the foundation for the subsequent development and application of viral vector tools for clinical trials [13].

Gene delivery technologies not only hold significant therapeutic implications for intracranial tumors but also offer new strategies for exploring novel anti-tumor mechanisms. For example, Liang et al. utilized AAV technology to intervene in the expression of insulin-like growth factor binding protein-2 (IGFBP2) in order to investigate its relationship with the development of brain gliomas. The results indicate that IGFBP2 can promote further development of brain gliomas by inducing M2-type macrophage polarization at the tumor site [112]. Chang et al. constructed an engineered AAV vector capable of expressing lipocalin-2 and thoroughly investigated the role of lipocalin-2 in the formation and development stages of medulloblastoma [113]. Cao et al. also utilized AAV-based gene delivery technologies to uncover a novel potential anti-glioma mechanism. Specifically, the authors engineered an AAV delivery system capable of expressing G protein inhibitory α subunit 2 (Gαi2) shRNA and found that inhibiting the expression of Gαi2 significantly suppressed the growth of glioma in the CNS [114]. Liu et al. proposed a potential novel therapeutic target for human glioma involving the inner mitochondrial membrane translocase 44 (TIMM44), which is a key protein in maintaining mitochondrial function. To validate this hypothesis, the authors constructed an AAV carrying TIMM44 shRNA. The experimental results showed that, after local injection, the growth of glioma cells in the brains of the model mice was significantly inhibited. The authors speculated that this may be due to the downregulation of TIMM44, leading to a decrease in ATP production in tumor cells and resulting in oxidative stress-induced cell apoptosis. In conclusion, overexpressed TIMM44 could be a novel and promising therapeutic target for human glioma [115]. The disruption of miRNA expression has been shown to be associated with the pathogenesis of various malignant intracranial tumors. Khalid Shah et al. first proposed that the simultaneous downregulation of miR-21 and upregulation of miR-7 can inhibit the growth of malignant gliomas. Specifically, the authors first designed and constructed an AAV delivery system capable of expressing miR-21 (miRzip-21). The experimental results demonstrate that by simultaneously modulating the upregulation of miR-21 and the downregulation of miR-7, the proportion of apoptosis in tumor cells significantly increased. Further in vivo experiments using mouse models of malignant brain tumors revealed that combined treatment with AAV-miRzip-21 and AAV-miR-7 could significantly inhibit tumor growth and prolong the survival of mice [116]. Chen et al. developed a hybrid genetic screening system and successfully screened for novel targets that enhance the anti-tumor activity of T cells using AAV vectors [117]. Chen et al. developed an AAV-based autochthonous genetic CRISPR screen for screening new targets in glioblastoma, paving the way for the development of gliomagenesis inhibitors [118].

In addition to the conventional AAV delivery system, other viral delivery systems have been developed for the treatment of intracranial tumors. For example, Hajitou et al. successfully constructed a hybrid AAV/phage (AAVP) for the treatment of glioblastoma by combining the genome of AAV with the M13 phage capsid that possesses the ability to incorporate the RGD4C ligand targeting αβ integrin receptors, achieving the dual targeting of the tumor [119]. Subsequently, the authors attempted to combine this engineered AAVP with temozolomide (TMZ), which enhanced transgene expression from RGD4C/AAVP in human GBM cells. The experimental results demonstrated that, following intravenous injection, RGD4C/AAVP efficiently targeted intracranial tumors and achieved the inhibition of glioblastoma growth through the synergistic action of TMZ and AAVP.

Overall, virus-based gene delivery technologies are widely and effectively utilized in the field of brain glioma treatment, showing broad prospects for development.

### 3.2. Integrating Virus-like Particle-Associated Therapy into Emerging Biotechnology Against Intracranial Cancer

Virus-like particles (VLPs) are artificially prepared biomimetic particles that mimic the structural characteristics of natural viruses but lack viral genomic material [120]. Unlike authentic viruses, VLPs are incapable of infecting cells due to their lack of essential genetic material. Similarly, serving as a common vector for gene delivery, VLPs possess inherent advantages over AAVs, particularly in terms of safety [121]. They are considered safer because they cannot replicate or infect host cells, but they can still elicit immune responses, which is beneficial for stimulating immunity while mitigating the risk of pathogenicity. Considering the adequate safety and the controllability of protein expression, VLPs are widely applied in the treatment of intracranial tumors [122]. Specifically, first, due to the absence of viral genomic material, VLPs do not trigger intracranial infections or other diseases, thereby reducing potential risks during therapy [123]. Second, VLPs typically exhibit high stability, allowing for storage and transportation under various environmental conditions, which is crucial for clinical applications. Additionally, VLPs can be engineered to encode various sequences of genes, enabling them to carry specific drugs, genes, or other therapeutic factors for the efficient treatment of intracranial tumors [124]. In summary, VLPs hold tremendous potential in the field of intracranial tumor therapy and are being extensively researched and developed for the treatment of various types of brain tumors. For example, Jiang et al. incorporated the natural retrovirus-like proteins found in the human brain into extracellular vesicles, allowing them to more efficiently protect nucleic acid drugs from RNase degradation and thereby transporting these drugs to intracranial lesions. This has a wide application prospect [125]. Steinmetz NF et al. discovered that plant-derived cowpea mosaic virus-like particles have the capability to activate immune responses within the intracranial glioma microenvironment and reshape the intracranial immune milieu [126]. To investigate this, the authors established a mouse model of brain glioma and demonstrated that CPMV immunotherapy effectively recruits innate immune cells as well as tumor-specific CD8^+^ T cells into the brain parenchyma within the glioma environment. Simultaneously, it reduces the population of immunosuppressive cells, leading to the inhibition of intracranial glioma [127]. Yang et al. devised a broccoli light-up three-way junction (b-3WJ) RNA scaffold capable of carrying silenced radiotherapy-resistant genes and encapsulated it within bacteriophage Qβ particles (TrQβ@b-3WJ). This was performed to achieve more precise RNA loading and ensure stable therapeutic effects while overcoming glioma cell resistance to radiotherapy. The experimental findings illustrate that TrQβ@b-3WJ effectively silenced epidermal growth factor receptor and IKKα, thus leading to the inactivation of the NF-κB signaling pathway and the inhibition of DNA repair. Compared to the control group, treatment with TrQβ@b-3WJ in conjunction with radiotherapy significantly prolonged the survival time of mice without evident systemic toxicity [128] (Figure 6).

Based on previous experimental studies, Sakamuro et al. found a significant upregulation of sterile alpha motif and HD domain-containing protein 1 (SAMHD1) in tumor tissues of glioblastoma patients, which is crucial for tumor initiation and progression. The authors designed and constructed VLPs carrying SAMHD1 depletion genes, which were then combined with γ-irradiation or temozolomide. The experimental results demonstrated that SAMHD1 serves as an ideal therapeutic target for brain glioma, improving the efficacy of temozolomide (TMZ) and γ-irradiation in glioblastoma (GBM) treatment [129]. Ren et al. developed a VLP with dual-targeting functions for treating glioblastoma. Specifically, the VLP was equipped with a brain-targeting peptide to traverse the BBB and utilized the tumor vascular preferred ligand RGD to selectively infect glioblastoma. Subsequently, the authors co-loaded paclitaxel and YAP (YAP overexpression promotes glioma cell migration and invasion) siRNA gene segments into the VLP. The experimental results showed that the designed VLP significantly enhanced drug accumulation in tumor tissues and achieved the inhibition of tumor cell growth and migration with minimal dosages [130]. Steinmetz et al. developed VLPs based on bluetongue virus for the delivery of herpes simplex virus 1 thymidine kinase (TK-VLPs). The experimental results demonstrated that the co-administration of TK-VLPs with ganciclovir synergistically inhibited the growth of glioblastoma cells [131]. Yang et al. loaded epirubicin inside VLPs, modified the surface with a cell-penetrating peptide, and labeled it with (68)Ga-DOTA, constructing a dual-functional drug delivery system for imaging and therapy, thus enabling the diagnosis and treatment of brain tumors [132]. In addition, based on the same strategy, Yang et al. construct a VLP modified with a cell-penetrating peptide and an apolipoprotein E peptide, which can downregulate the expression of the hepatocyte growth factor receptor gene. The results showed that this VLP could cross the BBB, inhibit the DNA repair mechanism, and synergistically enhance the anti-tumor efficacy with temozolomide [133]. Wang et al. designed a JCPYV-based VLP carrying the thymidine kinase suicide gene for glioblastoma multiforme treatment by utilizing the ability of neurogenic JC polyomavirus to infect glial cells. In combination with ganciclovir, the survival of mice was significantly prolonged, and the growth and metastasis of tumors were effectively inhibited [134].

### 3.3. Integrating Bacteria-Associated Therapy into Emerging Biotechnology Against Intracranial Cancer

Bacteria had long been viewed as a significant threat to human life and health until Dr. William Coley discovered that *Streptococcus pyogenes* could halt the growth of malignant tumors. Looking back on the history of bacterial development from today’s perspective, bacteria and their derivatives have played significant roles in the treatment of various diseases attributed to the intrinsic advantages of bacteria. First, bacteria have sufficient internal space and surface area to efficiently load various types of drugs [135]. Second, bacteria possess the characteristic of autonomous targeting, enabling them to actively seek out infection sites or tumor tissues within the body, thereby enhancing the efficiency of drug delivery [136]. In addition, bacteria and their derivatives themselves possess strong immunogenicity, capable of triggering immune responses, which is crucial to activating the body’s immune defense mechanisms to combat diseases, especially in cancer therapy [137]. In addition, certain bacteria have specific functions and are often used to treat certain diseases, for example, using bacteria with the ability to cross the BBB as drug carriers against CNS diseases [2].

One of the key challenges in intracranial cancer treatment is how to build efficient delivery systems that can cross the BBB. He et al. developed a complete bacterial delivery system capable of carrying nanoparticles against glioblastoma. In this research, glucose polymer and indocyanine green-modified silicon nanoparticles were loaded into co-anaerobic bacteria via the ABC transporter to form a Trojan bacterial drug carrier system [58]. The in vivo experimental results showed that this system exhibits higher brain accumulation compared to nanoparticles, with fluorescence predominantly concentrated in glioblastoma tissues, demonstrating the efficient targeted delivery and deep penetration of this delivery system into tumor tissues. Furthermore, under 808 nm laser irradiation, the indocyanine green loaded within the bacteria can generate a photothermal effect to destroy the bacterial carriers and kill tumor cells. Additionally, the release of damage-associated and pathogen-associated molecular patterns during this process aids in recruiting and infiltrating immune cells, thereby reshaping the suppressive immune microenvironment at the tumor site and activating systemic specific anti-tumor immunity. Based on previous reports that EC-K1 can cross the BBB and enter the intracranial space through a receptor-mediated pathway, Han et al. discovered that dead EC-K1 can also penetrate the BBB. Considering the significant advantage of dead EC-K1 retaining the intact structure and chemotaxis of live EC-K1 while losing its pathogenicity, the authors designed a drug delivery system based on dead *EC-K1* [138]. Specifically, the authors first co-incubated EC-K1 with therapeutic drugs modified with maltodextrin (MD), such as indocyanine green, allowing the drugs to enter the bacteria through the maltodextrin transporter pathway. Subsequently, the bacteria were inactivated by UV irradiation. The experimental results showed that EC-K1 could efficiently carry the drugs to penetrate the BBB and enter the intracranial space, and the mice maintained good health throughout the 14-day treatment period (Figure 7).

In addition to bacteria, bacterial derivatives such as bacterial outer membrane vesicles are also used as delivery carriers for intracranial tumor therapy. Compared to bacterial carriers, bacterial outer membrane vesicles possess non-replicability and lower immunogenicity while inheriting various functional proteins of the bacteria themselves, making them a widely used delivery vehicle for intracranial disease treatment drugs. For example, given that Gram-negative *Escherichia coli K1* can rely on outer membrane protein A to recognize gp96 (a special protein overexpressed on the surface of vascular endothelial cells) and cross the BBB, Han et al. tried to utilize bacterial outer membrane vesicles derived from *K1* as delivery vehicles for treating intracranial malignant tumors [18]. Concretely, first, the authors knocked out the lipopolysaccharide of K1 bacteria and collected lipopolysaccharide-free bacterial outer membrane vesicles (dMOVs) using a centrifugation method. Subsequently, PLGA nanoparticles were loaded into dMOVs (dMOVs@NPs) and intravenously injected. The experimental results showed that dMOVs@NPs efficiently reached the intracranial tumor lesions and effectively suppressed tumor growth. Additionally, the immunological results indicated that treatment with dMOVs@NPs could safely and effectively increase the levels of inflammatory cytokines within a safe range.

## 4. Emerging Microorganism-Based Therapy for Other CNS Diseases

### 4.1. Emerging Microorganism-Based Therapy for CNS Inflammation

The pathogenesis and characteristics of CNS inflammation have been described in detail in Section 2. The applications of microorganism-based therapies based on emerging biomedical technologies in CNS inflammation are summarized here.

In the field of CNS inflammation treatment, microorganism-based nucleic acid therapy represents a main strategy that has attracted much attention recently. For example, using virus-based gene editing technology, Ari Waisman et al. provided a novel therapeutic strategy for multiple sclerosis, a common chronic inflammation of the CNS. Concretely, the authors found that the lack of A20 in CNS endothelial cells, a ubiquitin-modifying protein that negatively regulates NF-κB signaling, would aggravate the experimental autoimmune encephalomyelitis (EAE) in mice. The molecular mechanism of A20 was further investigated, and it was found that the downregulation of A20 leads to intercellular adhesion molecule 1, vascular cell adhesion molecule 1, and the inducible T cell costimulator ligand (ICOSL) being significantly upregulated. Given these research results, the authors constructed a CNS-microvasculature endothelial cell, a specific AAV capable of silencing ICOSL to achieve the regulation of ICOSL. The results showed that after the AAV was injected into mice, some of the experimental severe autoimmune encephalomyelitis mouse models were significantly relieved, which provided an effective treatment strategy for the treatment of CNS inflammation [139].

In EAE patients, the expression level of glial glutamate transporters is downregulated, which makes neurons more vulnerable to the effects of glutamate excitotoxicity, causing various complications, such as retinal ganglion cell (RGC) degeneration and functional abnormalities. To further explore the mechanism of the effect of EAE on retinal ganglion cells, the authors made a reasonable guess that the expression levels of GLAST in the retina of rats are reduced due to the onset of EAE, which leads to the degeneration and functional abnormalities of retinal ganglion cells. Given this guess and relevant research, the authors constructed an AAV carrying GLAST mRNA to investigate whether the overexpression of GLAST in the retina could protect RGC against degeneration. The experimental results showed that the AAV increased the expression of GLAST in the retina and protected retinal ganglion cells from the effects of inflammatory CNS diseases [140]. Analogously, Hu et al. also used an AAV as the genetic vector to explore the important role of sterile α and TIR motif-containing protein 1 in optic neuritis induced by EAE [141].

As an important channel of CNS immune cell infiltration, the choroid plexus plays an important role in the maintenance of CNS homeostasis and the occurrence of diseases. For example, reports have shown that the onset and development of EAE are also related to the choroid plexus. To further investigate its critical role in CNS disorders, Lin et al. engineered an AAV to achieve the knockdown of relevant genes in choroid plexus tissue. The results showed that, after AAV injection into the lateral ventricle, the carried RNAi successfully silenced the expression of specific genes. Knocking down the adenosine A2A receptor using an AAV led to a significant alleviation of EAE symptoms in experimental mice [142].

C-X-C motif chemokine 12 represents a multifunctional chemokine. On the one hand, studies have shown that CXCL12 can increase the degree of CNS inflammation, and on the other hand, CXCL12s play a key role in promoting the recovery of EAE. Li et al. conducted a study to delve into the mechanisms of CXCL12 in the onset and treatment of EAE. Concretely, they constructed an AAV carrying the CXCL12 gene and injected it into the body through intrathecal catheterization to induce the upregulation of CXCL12 in the spinal cord. The experimental results demonstrated a significant upregulation of CXCL12 in the spinal cord after AAV injection, leading to a marked alleviation of the clinical signs and symptoms of EAE and a significant reduction in leukocyte infiltration associated with EAE. Furthermore, the authors proved that CXCL12 can promote the maturation and differentiation of oligodendrocyte precursor cells, which play a significant role in promoting remyelination in the spinal cord [143]. In addition, Leucine-rich repeat-containing protein 4 (LRRC4) is a key protein that regulates the formation of excitatory synapses and promotes axon differentiation, and mutations in LRRC4 can lead to CNS diseases such as autism spectrum disorder. Wu et al. utilized AAV to deliver the LRRC4 gene into the CNS of EAE-afflicted mice, thereby increasing the expression levels of LRRC4 protein in the spinal cords of the mice. The results showed that the ectopic expression of LRRC4 alleviated the clinical symptoms of EAE mice and protected neurons from immune damage [144].

Abnormal expression of interleukin-21 (IL-21) can induce a variety of autoimmune diseases. Espejo C et al. investigated the role of IL-21 in the pathogenesis of EAE by using mice as experimental models. Specifically, the authors introduced a gene encoding a novel soluble cytokine receptor (sIL21R) protein into an AAV to block the binding between IL-21 and its receptor (IL-21R). The results show that IL-21 plays a key role in the pathogenesis of EAE [145]. Analogously, the key roles of Nav1.6, IFN-α, p38MAPK/SGK1 signaling, interleukin-27, and sirtuin 1 in the occurrence and treatment of CNS inflammation were investigated recently with the help of a virus-based gene delivery system [146].

### 4.2. Emerging Microorganism-Based Therapy for Neurodegenerative Diseases

AAV-based gene drug delivery systems are of great significance for the exploration of new mechanisms of neurodegenerative diseases and the development of related emerging target inhibitors/agonists. Here, we introduce applications of AAV-based gene drug delivery systems for various neurodegenerative diseases over the last three years.

#### 4.2.1. Emerging Microorganism-Based Therapy for Parkinson’s Disease

Parkinson’s disease is a chronic progressive neurological disorder characterized by motor impairments, including muscle rigidity, tremors, bradykinesia, and postural instability. This disease is typically caused by the degeneration and death of dopaminergic neurons in the brain that control movement, leading to a decrease in dopamine levels. AAV-based gene therapy offers a novel strategy for the treatment of Parkinson’s disease. Lu et al. designed and constructed an engineered AAV that can selectively target striatal D1 medium spiny neurons. By loading promoter elements with enhanced D1-MSN activity into AAV, the treatment of Parkinson’s disease and the restoration of motor ability are realized [147] (Figure 8). Dawson VL et al. constructed an animal model of neurodegenerative disease with the overexpression of pathologic α-synuclein in AAV vectors to explore the pharmacological role of PAAN in neurodegenerative diseases [148]. MJ et al. introduced gene fragments with glutamic acid decarboxylase into AAV to investigate the safety and efficacy of subthalamic nucleus gene therapy for Parkinson’s disease [149]. Lanciego JL et al. used an AAV to construct an animal model of corticosynuclein disease for the end-stage preclinical screening of new drugs targeting alpha-synuclein [150]. Lie et al. investigated the role of the neurotoxic α-synuclein (α-syn) oligomers in the pathogenesis of Parkinson’s disease with an AAV [151]. Huang and colleagues developed an AAV capsid, Bl-hTFR1, which can bind to the human transferrin receptor, thereby facilitating AAV entry into the brain. The experimental results showed that, compared to traditional AAVs, the expression of the target gene in the brain was increased by 40–50 times, demonstrating promising therapeutic efficacy in the treatment of Parkinson’s disease [12].

#### 4.2.2. Emerging Microorganism-Based Therapy for Alzheimer’s Disease

Alzheimer’s disease is a progressive neurodegenerative disease and is the most common form of dementia in the elderly [152]. It is characterized by gradually progressive cognitive impairment and decline in memory as the primary features, ultimately leading to impairment in an individual’s daily living and social functioning. Similar to Parkinson’s disease, the discovery of mechanisms in Alzheimer’s disease and the development of new drugs benefit from AAV-based gene delivery strategies. For example, Campos et al. introduced shRNA of the RTP801 gene into an AAV to investigate the impact of RTP801 protein levels on the onset of Alzheimer’s disease. The experimental results showed that by using this engineered AAV to silence RTP801 expression in the hippocampal neurons of 5xFAD mice, cognitive deficits in the experimental animals were significantly improved. Furthermore, after the downregulation of RTP801 expression, the levels of glial markers associated with pathology and key inflammatory proteins returned to normal, demonstrating that RTP801 could be a potential target for future Alzheimer’s disease therapies [153].

Adiponectin (APN) is a type of adipokine that is expressed specifically in adipocytes and has neuroprotective and anti-inflammatory effects. To further explore the specific mechanism of APN in Alzheimer’s disease, Chan et al. constructed an AAV (AAV2/8-APN) expressing APN by inserting an APN gene fragment. The experimental results showed that the overexpression of APN decreased both the soluble and fibrillar Aβ in the brains of 5xFAD mice and reduced the secretion of IL-1β and IL-18. Subsequent experimental results further showed that treatment with AAV-APN significantly improved memory function and reduced atrophic neuronal processes in 5xFAD mice [154].

In Alzheimer’s disease (AD), triggering receptor expressed on myeloid cell 2 (TREM2) has been shown to have neuroprotective effects against inflammation and nerve damage. Based on this, Wang et al. introduced TREM2-silencing genes into an AAV to explore the effect of reduced TREM2 expression on Alzheimer’s disease. The experimental results showed that after TREM2 was silenced, the level of pro-inflammatory cytokines in the brain of mice increased inflammation and significantly decreased cognitive levels. In contrast, the overexpression of TREM2 mitigated LPS-induced inflammation in vitro and induced M2-phenotypic microglia [155].

### 4.3. Emerging Microorganism-Based Therapy for Stroke

Stroke is a CNS disease with a high incidence, causing great harm. Recently, the development of a number of new treatment strategies for stroke has attracted much attention. Among them, the virus-based gene delivery system represents a mainstream strategy for stroke treatment.

For example, vascular endothelial growth factor-C (VEGF-C) can impact the stability of the CNS by regulating the development of meningeal lymphatic vessels. Based on this, Thomas JL et al. further explored the effects of VEGF-C overexpression on the onset of ischemic stroke. Specifically, the authors engineered an AAV carrying a segment of mRNA encoding VEGF-C (AAV-mVEGF-C), which was intrathecally injected into a mouse model of stroke. The experimental results showed that AAV-mVEGF-C pretreatment reduced stroke injury and improved motor performance in the subacute stage, possibly attributed to the neuroprotective effects of VEGF-C [156].

It has been reported that the knockout of polypyrimidine tract-binding protein 1 (PTB) can reprogram astrocytes into functional neurons. Therefore, Huang et al. introduced PTB-silencing gene fragments into an AAV to knock out PTB in mice with ischemic stroke induced by endothelin-1 and then investigated the therapeutic effect on stroke. The results showed that in a mouse model of ischemic stroke, the knockout of PTB effectively transformed endogenous glial cells into functional neurons, thereby restoring the neural tissue structure at the ischemic site. In addition, the elimination of PTB reduced the inflammatory response at the lesion and improved the behavioral function of the stroke mice. In conclusion, this study provides a new strategy for the treatment of stroke [157]. Similarly, Gao et al. conducted an in-depth exploration of the pathogenesis of ischemic stroke. The overexpression of microglial PU.1 in an AAV vector successfully reversed the protective effect of HDAC3 knockout on ischemic stroke, demonstrating the key pathological role of the HDAC3/PU.1 axis in the pathogenesis of this condition [158] (Table 3).

## 5. Concluding Remarks

In the past decade, considerable efforts have been made to develop smart drug delivery systems against CNS diseases by integrating the emerging biomedical discovery and technological innovation with traditional microorganism therapy. In this review, we closely focused on the applications of microorganism-based emerging biotechnology therapy in the treatment of CNS diseases in the past three years. First, we explored the necessary relationship between microorganisms and the occurrence and development of CNS diseases. Second, we detailed the latest preclinical studies of various microbial therapies (such as bacteriotherapy, bacterial outer membrane vesicle vectors, OVs, AAVs, and VLPs) in the treatment of several common CNS diseases, including brain cancer, CNS inflammation, neurodegenerative diseases, and stroke. In general, microorganism-based emerging biomedical therapy aids in the treatment of CNS diseases, the discovery of new therapeutic targets, and the development of new drugs.

## 6. Clinical Challenges

Despite the potential benefits of microorganism-based emerging biomedical therapy, there exist several challenges that hamper clinical development in the field of CNS treatment. First, the CNS is a very fragile immune organ; so, it is crucial to achieve disease treatment without affecting CNS homeostasis. Second, although microbial-related delivery systems have a certain ability to leap over the BBB compared to other delivery vectors, systematic delivery strategies of microorganisms are difficult to implement due to their inherent immunogenicity, thus affecting the effectiveness of intracranial targeting. As reviewed in this paper, although there have been studies aimed at improving the long-term cycling capacity of microorganisms through gene editing techniques (such as knocking out lipopolysaccharides of K1 bacteria), many have been limited to invasive drug delivery methods. In addition, the treatment of CNS diseases is associated with higher costs owing to the complex structure. This includes expenses during the drug development phase as well as the financial burden on patients during clinical treatment. These factors significantly hinder the clinical application of microbiome-based drugs in the treatment of CNS diseases. The significance of this review lies in making relevant research more clearly understood and accepted by the public, thereby promoting the progress of clinical development as much as possible. Overall, in this review, we have summarized the preclinical applications of microorganism-based emerging biomedical therapy in the treatment of CNS diseases and laid a foundation for the subsequent development of related clinical products. With concerted efforts in addressing these challenges, microorganism-based emerging biomedical therapy has the potential to be applied in clinical settings for the treatment of CNS diseases (Figure 9).

## Figures and Tables

**Figure 1 pharmaceutics-17-01175-f001:**
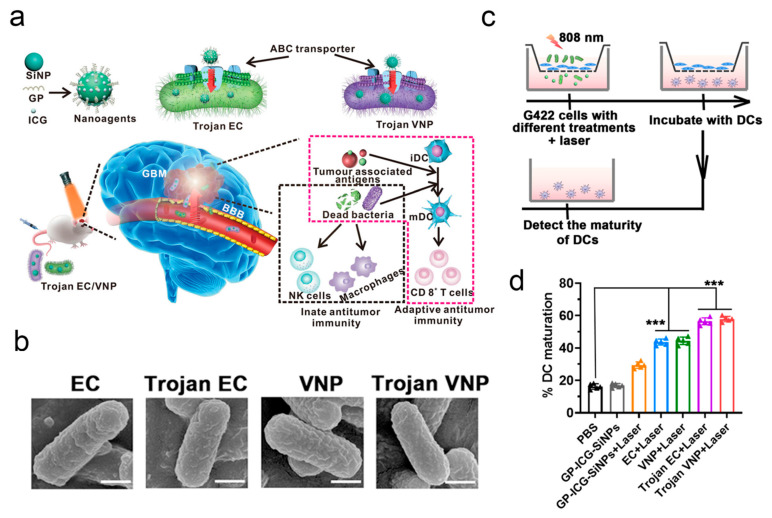
Bacteria loaded with glucose polymer and photosensitive ICG silicon nanoparticles for glioblastoma photothermal immunotherapy, *** *p* < 0.001 by two-tailed Student’s *t*-test. (**a**) Schematic illustration of Trojan bacteria crossing the blood-brain barrier for photothermal immunotherapy of glioblastoma. (**b**) SEM images of EC, Trojan EC, VNP, and Trojan VNP. Scale bars: 200 nm. (**c**) Schematic diagram of the transwell system. G422 cells were cultured in the upper chamber, and DCs were cultured in the lower chamber. (**d**) Quantification of the maturation of DCs after different treatments, as indicated in the transwell system. Reprinted with permission from Ref. [58].

**Figure 2 pharmaceutics-17-01175-f002:**
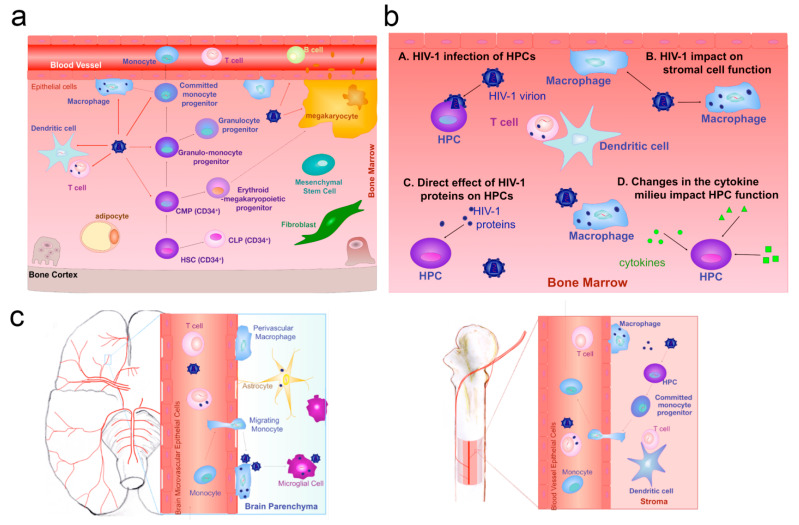
HIV-1 infection of bone marrow hematopoietic progenitor cells and their role in trafficking and viral dissemination. (**a**) Cells of the BM susceptible to HIV-1 infection. (**b**) Cells of the BM susceptible to HIV-1 infection. (**c**) Trafficking of HIV-1-infected cells from the BM to the CNS. Reprinted with permission from Ref. [71].

**Figure 3 pharmaceutics-17-01175-f003:**
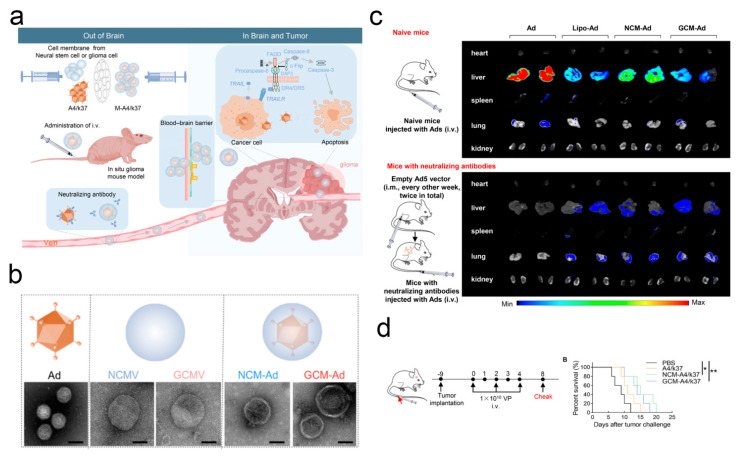
Cell membrane-coated oncolytic adenovirus for targeted treatment of glioblastoma, * *p* < 0.05, ** *p* < 0.01 by two-tailed Student’s *t*-test. (**a**) Design and mechanism of a cell membrane-coated oncolytic adenovirus for GBM treatment. (**b**) TEM images of Ad (CRAd5), NCMV, GCMV, NCM-Ad, and GCM-Ad negatively stained with uranyl acetate. The scale bar is 100 nm. (**c**) Images of virus-mediated RFP expression in different tissues from naïve and preexisting immunity mouse models (n = 3). (**d**) Experimental procedure for evaluating the anti-tumor effects of M-A4/k37 in vivo. Survival curve of each group after treatment (n = 5). Reprinted with permission from Ref. [94].

**Figure 4 pharmaceutics-17-01175-f004:**
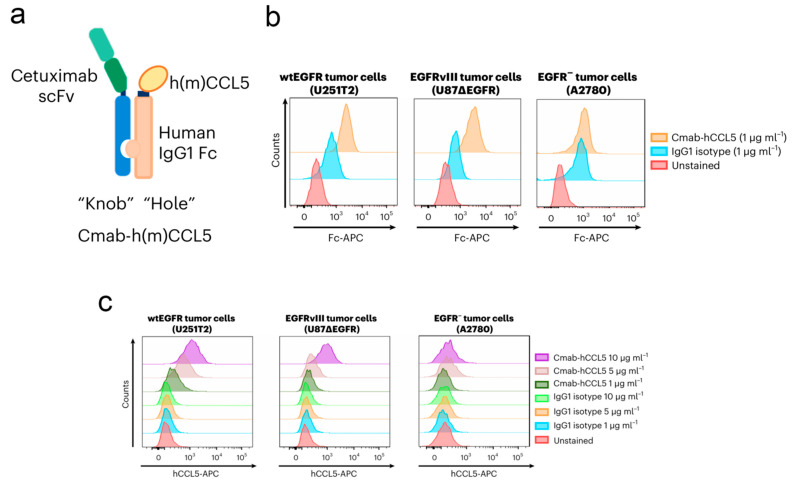
Specific targeting of glioblastoma with an oncolytic virus expressing a cetuximab-CCL5 fusion protein via innate and adaptive immunity. (**a**) Schematic of the Cmab-h(m)CCL5 bispecific fusion protein. scFv: single-chain variable fragment. (**b**) Detection of purified Cmab-hCCL5 bound to the wild-type EGFR (wtEGFR) U251T2 and EGFRvIII U87ΔEGFR GBM cell lines or the EGFR-A2780 human ovarian cancer cell line, as measured by flow cytometry after staining Cmab-hCCL5-incubated tumor cells with anti-Fc-allophycocyanin (APC) (**b**) or anti-hCCL5-APC (**c**). Cmab-hCCL5 was purified from lentivirus-infected CHO cells. Reprinted with permission from Ref. [98].

**Figure 5 pharmaceutics-17-01175-f005:**
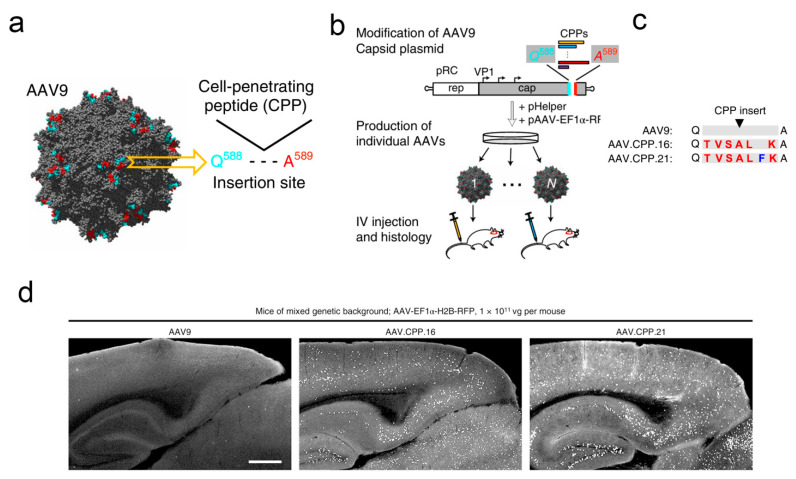
Variants of the adeno-associated virus serotype 9 with enhanced penetration of the blood–brain barrier in rodents and primates. (**a**) AAV9 capsid model showing the insertion site for CPPs. (**b**) Schematic diagram showing the AAV screening protocol. (**c**) Sequences of peptide inserts on AAV.CPP.16 and AAV.CPP.21 capsids. (**d**) Representative images of brain regions in mice of mixed genetic background showing RFP-labeled cells (white dots) 21 days after intravenous (IV) administration. Reprinted with permission from Ref. [111].

**Figure 6 pharmaceutics-17-01175-f006:**
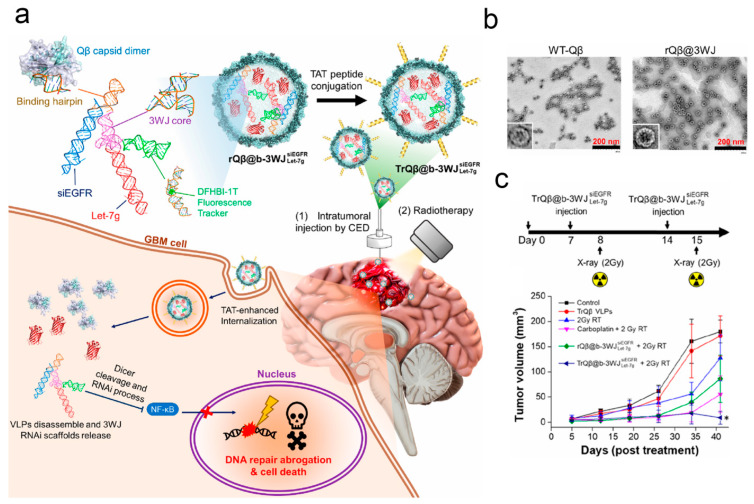
Bioengineered bacteriophage-like nanoparticles as RNAi therapeutics to enhance radiotherapy against glioblastomas, * *p* < 0.05 by two-tailed Student’s *t*-test. (**a**) Schematic illustration of the effective tumor radiotherapy, enhancing RNAi penetration and inhibiting DNA repair in GBMs by TrQβ@b-3WJ_Let-7g_ using CED infusion. (**b**) Transmission electron microscopy (TEM) image of WT-Qβ and rQβ@b-3WJ MG. The scale bar represents 200 nm. (**c**) Treatment protocols assessing multigene silencing-enhanced radiotherapy. Reprinted with permission from Ref. [128].

**Figure 7 pharmaceutics-17-01175-f007:**
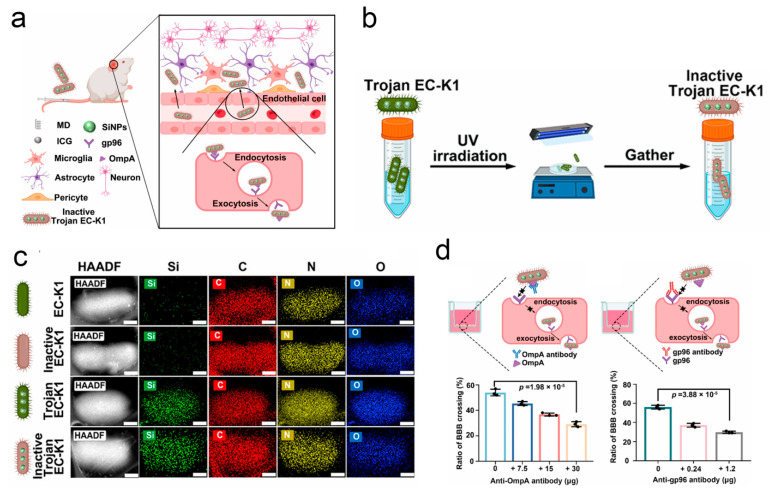
Inactive Trojan bacteria as safe drug delivery vehicles crossing the BBB. (**a**) Schematic illustration of inactive Trojan EC-K1 crossing the BBB. (**b**) Schematic illustrating the construction of the inactive Trojan EC-K1 system. (**c**) Elemental mapping in HAADF-STEM images of EC-K1, inactive EC K1, Trojan EC-K1, and inactive Trojan EC-K1. Scale bars: 200 nm. (**d**) Schematic illustrating the competition experiments to confirm the uptake of inactive Trojan EC-K1 by bEnd.3 cells associated with OmpA and the corresponding penetration rates of inactive Trojan EC-K1 in the presence of the OmpA antibody, along with the corresponding penetration rates of inactive Trojan EC-K1 in the presence of the gp96 antibody. Reprinted with permission from Ref. [138].

**Figure 8 pharmaceutics-17-01175-f008:**
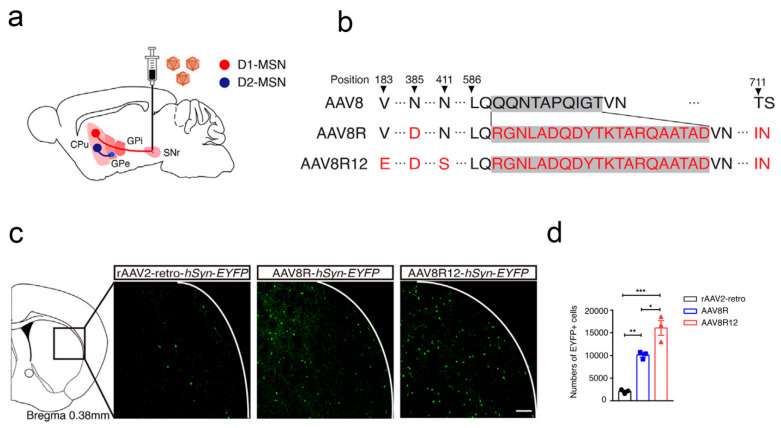
Circuit-specific gene therapy reverses core symptoms in a primate Parkinson’s disease model. * *p* < 0.05, ** *p* < 0.01 and *** *p* < 0.001 by two-tailed Student’s *t*-test. (**a**) Schematic illustration of inactive Trojan EC-K1 crossing the BBB. (**b**) Schematic illustrating the construction of the inactive Trojan EC-K1 system. Mutations were introduced at three sites in the AAV8 capsid protein to make AAV8R.Two additional mutations of V183E and N411S were in corporated into the AAV8R12 capsid protein. (**c**) Elemental mapping in HAADF-STEM images of EC-K1, inactive EC-K1, Trojan EC-K1, and inactive Trojan EC-K1. Scale bars: 200 nm. (**d**) Schematic illustrating the competition experiments to confirm the uptake of inactive Trojan EC-K1 by bEnd.3 cells associated with OmpA and the corresponding penetration rates of inactive Trojan EC-K1 in the presence of the OmpA antibody, along with the corresponding penetration rates of inactive Trojan EC-K1 in the presence of the gp96 antibody. Reprinted with permission from Ref. [147].

**Figure 9 pharmaceutics-17-01175-f009:**
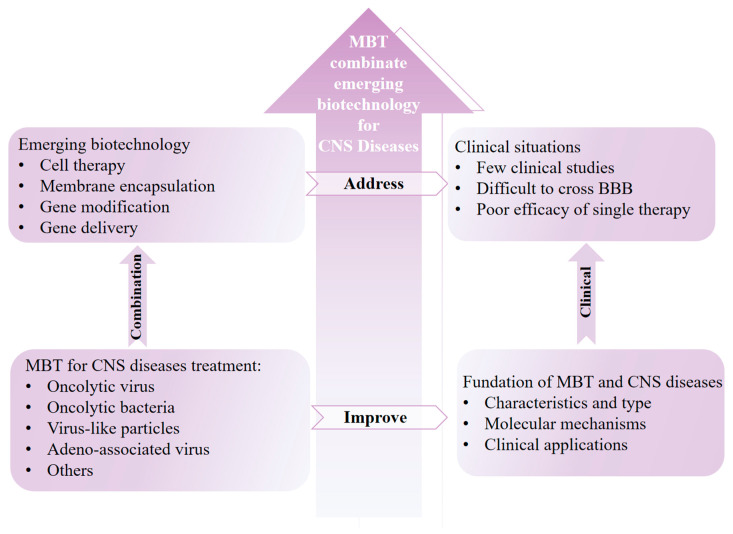
The clinical situations and future prospects of microorganism-based therapy in emerging biotechnology against intracranial central nervous system diseases.

**Table 1 pharmaceutics-17-01175-t001:** Cross-talk between CNS diseases and microorganisms.

Microbial Type	Disease Type	Entry Mode	Concrete Pathway	Reference
EC-K1	suppurative meningitis	receptor-ligand binding	OmpA-gp96	[47]
EC-K1	suppurative meningitis	receptor-ligand binding	CNF1 protein	[48]
*Streptococcus pneumoniae* *group B*, *Streptococcus* *neonatal meningitis*, and *Escherichia coli*	bacterial meningitis and cerebral palsy	hijacking the iron transporter receptor	transporter receptor	[49]
*Streptococcus pneumoniae*	bacterial meningitis	hijacking the polymeric immunoglobulin receptor	polymeric immunoglobulin receptor	[50]
*S. agalactiae*	bacterial meningitis	receptor-ligand binding	Lmb, FbsA, and IagA	[51]
*L. monocytogenes*	bacterial meningitis	receptor-ligand binding	InlB	[52]
*S. pneumoniae*	bacterial meningitis	receptor-ligand binding	phosphorylcholine and PAF receptor	[53]
*N. meningitidis*	bacterial meningitis	receptor-ligand binding	protein Opc-fibronectin	[54]
*M. tuberculosis*	tuberculous meningitis	hijacking immune cells	scavenger receptors	[55]
*S. pneumoniae*	bacterial meningitis	increasing TNF-α level	phosphorylcholine relative receptor	[56]
Reovirus	viral meningitis	receptor-ligand binding	JAM-A	[62]
Retrovirus HTLV1	viral meningitis	receptor-ligand binding	GLUT	[63]
SARS-CoV-2	viral meningitis	upregulation of chemokines	upregulation of the MMP9 protein	[64]
Polyomavirus	viral meningitis	receptor-ligand binding	serotonin receptor 2A	[65]
Mouse adenovirus type 1	acute encephalomyelitis	BBB destruction	BBB-destroying E3 protein	[65]
West Nile virus	encephalitis	BBB destruction	MMP-9	[67]
Poliovirus	encephalitis	receptor-ligand binding	CD155	[68]
Adenovirus	encephalitis	receptor-ligand binding	CAR	[68]
Rabies virus	encephalitis	receptor-ligand binding	NCAM	[68]
Herpes virus and pseudorabies virus	encephalitis	receptor-ligand binding	PVRL1, 2	[69]
Trojan horse	encephalitis	Trojan horse	infection of monocytes and macrophages	[69]
Chlamydia pneumoniae	Alzheimer’s disease	through the nasal nerves	nasal nerves	[74]

**Table 2 pharmaceutics-17-01175-t002:** Applications of boarding OVs onto biomedical technologies against intracranial cancer in clinical trials [103].

Country	Type of OV	Years	Administration Route	Biomedical Technologies	Clinical Stage	Type of Disease	References
USA	Ad (DNX-24-RGD)	2016	i.t.	Combined with pembrolizumab	II	Malignant brain tumor	NCT02798406
USA	HSV (G207)	2023	MRI-guided infusion	Gene modification	II	High-grade glioma	NCT04482933
USA	PVSRIPO	2017	i.t.	Gene modification	II	Malignant glioma	NCT02986178
Japan	HSV (G47 delta)	2014	i.t.	Gene modification	II	Glioblastoma	UMIN000015995
USA	HSV (M032)	2022	Infusion	Gene modification	II	Glioblastoma multiforme	NCT05084430
China	HSV (OH2)	2021	Ommaya reservoir injection	Gene modification	II	CNS tumor	NCT05235074
Germany	H-1 parvovirus	2011	i.t./i.v.	Gene modification	II	Glioblastoma multiforme	NCT01301430
France	TG6002	2017	i.v.	Gene modification	II	Glioblastoma	NCT03294486
The Netherlands	Ad (DNX-24-RGD)	2010	Convection-enhanced delivery	/	II	Recurrent glioblastoma	NCT01582516
Spain	Ad (ICOVIR-5)	2021	Infusion	Stem cell therapy	II	Medulloblastoma	NCT04758533
USA	PVSRIPO	2017	Convection-enhanced delivery	/	Ib	Malignant glioma	NCT03043391
USA	HSV (G207)	2019	MRI-guided infusion	Gene modification	I	Glioblastoma multiforme	NCT03911388
USA	REOLYSIN^®^	2006	i.t.	/	I	Malignant glioma	NCT00528684
China	Ad-TD-nsIL12	2023	i.t.	Gene modification	I	DIPG	NCT05717712
Spain	DNX-24-RGD	2017	Cerebellar peduncle infusion	Gene modification	I	Brainstem glioma	NCT03178032
USA	DNX-24-RGD	2019	Infusion	Stem cell therapy	I	High-grade glioma	NCT03896568
USA	DNX-24-RGD	2014	i.t.	Combined with IFNγ	I	Recurrent glioblastoma	NCT02197169
Spain	DNX-24-RGD	2013	i.t. or resected cavity	Combined with temozolomide	I	Recurrent glioblastoma	NCT01956734
USA	CRAd-S-pk7	2017	Resected cavity	Stem cell therapy	I	High-grade glioma	NCT03072134
USA	CRAd-S-pk7	2023	i.c.	/	I	Recurrent high-grade glioma	NCT05139056
Spain	DNX-2440	2018	i.t.	/	I	Recurrent glioblastoma	NCT03714334
USA	Ad5-yCD	2022	i.t.	Gene modification	I	Malignant glioma	NCT05686798
USA	HSV (G207)	2019	Infusion	Gene modification	I	Glioblastoma multiforme	NCT03911388
USA	rQNestin	2017	i.t.	Gene modification	I	Malignant glioma	NCT03152318
USA	HSV (C134)	2019	i.t.	/	I	Recurrent glioblastoma	NCT03657576
USA	HSV (M032)	2013	Infusion	Gene modification	I	Recurrent glioblastoma	NCT02062827
USA	HSV (MVR-C5252)	2023	i.t.	Gene modification	I	Recurrent glioblastoma	NCT05095441
USA	PVSRIPO	2012	i.t.	/	I	Glioblastoma	NCT01491893

**Table 3 pharmaceutics-17-01175-t003:** Emerging microorganism-based therapy for intracranial cancer treatment.

Microbial Type	Disease Type	Emerging Biotechnology	Specific Empowerment	Reference
CRAd-S-pk7	Intracranial glioma	NSC therapy	Tumor tropism	[17]
OVs	Intracranial glioma	NSC therapy	Crossing the BBB, tumor targeting	[88]
Δ24-RGD	Intracranial glioma	MSC therapy	Crossing the BBB, tumor targeting	[90]
oHSV	Melanoma brain metastases	MSC therapy	Crossing the BBB, tumor targeting	[16]
OVs	Glioblastoma	CAR-T therapy	Enhancing the anti-tumor efficacy	[91]
A4/k37	Glioblastoma	Membrane encapsulation	Crossing the BBB	[94]
Ads	Glioblastoma	Gene modification	Enhancing the anti-tumor efficacy	[96]
OVs	Glioblastoma	Gene modification	Enhancing the anti-tumor efficacy	[97]
OVs	Glioblastoma	Gene modification	Enhancing the anti-tumor efficacy	[98]
OVs	Glioblastoma	Gene modification	Enhancing OV replication	[99]
Delta-24-ACT	Glioblastoma	Gene modification	Enhancing the anti-tumor efficacy	[100]
DNX-2401	DIPG	Combination with radiotherapy	Enhancing the anti-tumor efficacy	[101]
OVs	Glioblastoma	Combination with CPA	Inhibition of antiviral response	[102]
AAV-LIGHT	Glioblastoma	Gene delivery (LIGHT)	Enhancing the anti-tumor efficacy	[107]
HER2-AAV	Glioblastoma	Gene delivery (aPD-1)	Enhancing the anti-tumor efficacy	[109]
AAV	Glioblastoma	Gene delivery (IFN-β)	Tumor targeting, enhancing the anti-tumor efficacy	[110]
AAV.CPP.16	Glioblastoma	Gene delivery (membrane-penetrating peptide)	Crossing the BBB	[13]
AAV	Glioblastoma	Gene delivery (IGFBP2)	Exploring novel anti-tumor mechanisms	[112]
AAV	Medulloblastoma	Gene delivery (lipocalin-2)	Exploring novel anti-tumor mechanisms	[113]
AAV	Glioblastoma	Gene delivery (Gαi2)	Exploring novel anti-tumor mechanisms	[114]
AAV	Glioblastoma	Gene delivery (TIMM44)	Exploring novel anti-tumor mechanisms	[115]
AAV	Glioblastoma	Gene delivery (miRzip-21)	Exploring novel anti-tumor mechanisms	[116]
Hybrid AAV/phage	Glioblastoma	Gene delivery (RGD4C)	Enhancing the anti-tumor efficacy	[119]
Bacteriophage Qβ	Glioblastoma	Gene delivery (b-3WJ)	Enhancing the anti-tumor efficacy	[127]
VLPs	Glioblastoma	Gene delivery (SAMHD1)	Enhancing the anti-tumor efficacy	[129]
VLPs	Glioblastoma	Gene delivery (YAP)	Crossing the BBB, tumor targeting	[130]
VLPs	Glioblastoma	Gene delivery (TK-VLPs)	Enhancing the anti-tumor efficacy	[131]
VLPs	Glioblastoma	Gene delivery (Ga-DOTA)	Enhancing the anti-tumor efficacy	[131]
VLPs	Glioblastoma	Gene delivery (apolipoprotein E peptide)	Crossing the BBB, enhancing the anti-tumor efficacy	[133]
VLPs	Glioblastoma	Gene delivery (JCPYV)	Crossing the BBB, enhancing the anti-tumor efficacy	[134]
Bacteria	Glioblastoma	Combination with silicon nanoparticles	Crossing the BBB	[58]
EC-K1	Glioblastoma	Combination with photodynamic therapy	Crossing the BBB	[138]
EC-K1 OMVs	Glioblastoma	Combination with PLGA nanoparticles	Crossing the BBB	[18]
AAV	Multiple sclerosis	Gene delivery (ICOSL)	Enhancing efficacy	[139]
AAV	Encephalomyelitis	Gene delivery (GLAST)	Enhancing efficacy	[140]
AAV	Encephalomyelitis	Gene delivery (sterile α)	Enhancing efficacy	[141]
AAV	Encephalomyelitis	Gene delivery (A2A)	Enhancing efficacy	[142]
AAV	Encephalomyelitis	Gene delivery (CXCL12)	Enhancing efficacy	[143]
AAV	Encephalomyelitis	Gene delivery (LRRC4)	Enhancing efficacy	[144]
AAV	Encephalomyelitis	Gene delivery (IL-21)	Enhancing efficacy	[145]
AAV	Parkinson’s disease	Gene delivery (D1-MSN)	Enhancing efficacy	[147]
AAV	Parkinson’s disease	Gene delivery (pathologic α-synuclein)	Enhancing efficacy	[148]
AAV	Parkinson’s disease	Gene delivery (glutamic acid decarboxylase)	Enhancing efficacy	[149]
AAV	Parkinson’s disease	Gene delivery (α-synuclein)	Exploring novel mechanisms	[150]
AAV	Parkinson’s disease	Gene delivery (IL-21)	Exploring novel mechanisms	[151]
AAV	Parkinson’s disease	Gene delivery (Bl-hTFR1)	Exploring novel mechanisms	[12]
AAV	Alzheimer’s disease	Gene delivery (RTP801)	Exploring novel mechanisms	[153]
AAV2/8-APN	Alzheimer’s disease	Gene delivery (APN)	Enhancing efficacy	[154]
AAV	Alzheimer’s disease	Gene delivery (TREM2)	Enhancing efficacy	[155]
AAV-mVEGF-C	Stroke	Gene delivery (VEGF-C)	Enhancing efficacy	[156]
AAV	Stroke	Gene delivery (PTB)	Enhancing efficacy	[157]
AAV	Stroke	Gene delivery (HDAC3/PU.1)	Enhancing efficacy	[158]

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
