# Peer review of "Integrating Microorganism-Based Therapy and Emerging Biotechnology in the Treatment of Intracranial Central Nervous System Diseases"

_pharmaceutics, 2025, doi:10.3390/pharmaceutics17091175_

Round 1
Reviewer 1 Report
Comments and Suggestions for Authors
The manuscript by Li et al., reviews the microorganism-based drug delivery systems integrating emerging biotechnologies such as OVs, AAVs, VLPs, and bacterial vesicles in a broad spectrum of CNS diseases including glioblastoma, neuroinflammation, neurodegeneration, and stroke. It is well-researched and includes detailed insights into preclinical studies, mechanisms of action, and clinical trials. However, the manuscript is dense and could benefit from improved organization, clarity.
Here are some suggestions:
- The authors should consider consolidating overlapping content for conciseness and avoid repetition of concepts. For examples, the last sentence is the abstract is too long and repeats itself, …etc. More importantly, the manuscript should be extensively proofread for grammar and readability.
- The titles on the sections and subsections should be more concise and descriptive to improve and ensure smooth transitions between sections. For example:
- section 2.2 talks about outline of CNS diseases and then describe in the different subsections the CNS diseases. From the title it is difficult to understand what the authors intend to convey in this section and how it is different from information in 2.3
- Section 3 is overwhelming. The authors should consider focusing this section on the different delivery technologies e.g cell-based, membrane encapsulation, gene modification, and other biomedical technologies. Then describing within each technology the different microorganisms. Also, highlight the most promising therapies and technologies rather than presenting all studies equally.
Reviewer 2 Report
Comments and Suggestions for Authors
In this review, Li et al. provide an excellent overview of biotechnology applications related to CNS diseases.
However, some of the discussion raises concerns. For example, in Section 3.2, they state, "They are considered safer because they cannot replicate or infect host cells, thereby circumventing potential immune responses and mitigating the risk of pathogenicity" and "Furthermore, compared to traditional viral vectors, VLPs are more readily accepted by the immune system, reducing the likelihood of immune rejection and thereby enhancing treatment efficacy and duration." VLPs do not, in fact, fail to elicit immune responses; on the contrary, they are used in many vaccine applications.
Also, the section numbering should be reviewed. For example, "2.1. CNS barriers and anatomy," "2.2. Outline of CNS diseases" is followed by "2.1.1. Intracranial cancer." This section numbering seems illogical. In addition, the "2.2.2. CNS Inflammation" section lacks a detailed description of the underlying mechanisms. We recommend adding more information on the molecular biology of CNS inflammation to better understand the underlying principles.
Furthermore, a common issue across many sections is the lack of discussion of case studies (e.g., a review of the causes of clinical trial failure) and cost-effectiveness. Since cancer treatment is often associated with cost, we recommend adding a separate section to address this issue.
